# Current Landscape and Potential Challenges of Immune Checkpoint Inhibitors in Microsatellite Stable Metastatic Colorectal Carcinoma

**DOI:** 10.3390/cancers15030863

**Published:** 2023-01-30

**Authors:** María San-Román-Gil, Javier Torres-Jiménez, Javier Pozas, Jorge Esteban-Villarrubia, Víctor Albarrán-Fernández, Pablo Álvarez-Ballesteros, Jesús Chamorro-Pérez, Diana Rosero-Rodríguez, Inmaculada Orejana-Martín, Íñigo Martínez-Delfrade, Pablo Reguera-Puertas, Raquel Fuentes-Mateos, Reyes Ferreiro-Monteagudo

**Affiliations:** 1Medical Oncology Department, Ramón y Cajal University Hospital, 28034 Madrid, Spain; 2Medical Oncology Department, Clínico San Carlos University Hospital, 28040 Madrid, Spain; 3Medical Oncology Department, 12 de Octubre University Hospital, 28041 Madrid, Spain

**Keywords:** metastatic colorectal carcinoma (mCRC), pMMR, MSS, immune checkpoint inhibitors (ICI), biomarkers, POLE/POLD1, TMB, microbiome, immunoscore

## Abstract

**Simple Summary:**

The implementation of immunotherapy in the therapeutic landscape of colorectal carcinoma has been difficult due to the inefficacy reported in early clinical trials. Nevertheless, a small subset of tumors deficient in DNA mismatch repair proteins show excellent responses to immune checkpoint inhibitors, with long-lasting results. Most of our patients do not have these alterations and, therefore, immunotherapy appears not to be effective. In the current review, we attempt to describe the main mechanisms of resistance to immunotherapy presented by these tumors, how to cope with them, and the potential use of other biomarkers of response to immunotherapy.

**Abstract:**

Colorectal cancer (CRC) is the third most frequent cancer and the second most common cause of cancer-related death in Europe. High microsatellite instability (MSI-H) due to a deficient DNA mismatch repair (dMMR) system can be found in 5% of metastatic CRC (mCRC) and has been established as a biomarker of response to immunotherapy in these tumors. Therefore, immune checkpoint inhibitors (ICIs) in mCRC with these characteristics were evaluated with results showing remarkable response rates and durations of response. The majority of mCRC cases have high levels of DNA mismatch repair proteins (pMMR) with consequent microsatellite stability or low instability (MSS or MSI-low), associated with an inherent resistance to ICIs. This review aims to provide a comprehensive analysis of the possible approaches to overcome the mechanisms of resistance and evaluates potential biomarkers to establish the role of ICIs in pMMR/MSS/MSI-L (MSS) mCRC.

## 1. Introduction

According to the latest GLOBOCAN data, colorectal cancer (CRC) is the third most frequent cancer in terms of incidence rate in adults and the second most common cause of cancer-related deaths [1,2,3,4]. Approximately 25% of CRC patients are metastatic at diagnosis, and almost half of the patients with early-stage CRC will develop metastatic disease throughout their life. 

Most patients with metastatic colorectal cancer (mCRC) have an incurable disease, and treatment is based on systemic therapy with a palliative purpose. The prognosis for patients with mCRC remains poor, with an overall survival (OS) of approximately 30 months (m) [5]. 

Several target molecular biomarkers have changed the landscape of treatment of mCRC. KRAS and NRAS mutations are associated with primary resistance to anti-EGFR therapies, so cetuximab and panitumumab are only indicated for RAS wild-type mCRC. Other targeted therapies for mCRC that have the approval of the regulatory agencies are encorafenib and cetuximab in v-raf murine sarcoma viral oncogene homolog B1 (BRAF) V600E mutations, and larotrectinib or entrectinib in NTRK fusions [6].

Immune checkpoint inhibitors (ICI) have changed the prognosis of several cancers, leading to improvements in survival along with remarkable responses [7]. In mCRC, ICIs have demonstrated their efficacy for the treatment of tumors that are mismatch-repair-deficient (dMMR) with high levels of microsatellite instability (MSI-H) (termed dMMR/MSI-H or MSI tumors) [8,9]. By contrast, ICIs are ineffective in tumors that are mismatch-repair-proficient (pMMR) and are microsatellite-stable (MSS) or have low levels of microsatellite instability (MSI-L) (termed pMMR/MSS/MSI-L or pMMR/non-MSI-H tumors, also known as MSS) [8]. However, fewer than 5% of all mCRCs are MSI, with higher rates presented in earlier stages [10]. Therefore, combinations of chemotherapeutic (CT) agents with anti-vascular endothelial growth factor (VEGF) or anti-epidermal growth factor receptor (EGFR), depending on the RAS status, are still the standard treatment on first and successive lines in MSS mCRC [6,11]. For tumors with BRAFV600E mutation, the combination of encorafenib with cetuximab has been proved successful. [12,13].

Overall, this article aims to analyze the possible mechanisms of resistance to immunotherapy presented by MSS mCRC and how to overcome them. Additionally, we describe potential biomarkers of response to ICIs for consideration in this subgroup of patients.

## 2. Current Landscape of ICIs in MSI mCRC

### 2.1. Rationale for the Use of ICIs

As mentioned before, mCRC is subclassified into two groups based on mutation patterns and ability to repair DNA microsatellite damage: tumors with an MSI signature that generally has a higher mutation burden (>12 mutations per 106 DNA bases) and tumors with an MSS signature commonly expressing a lower mutation burden (<8.24 mutations per 106 DNA bases) [14].

The development of MSI tumors is a sporadic event in 70–85% of patients with mCRC, but MSI-H is the hallmark of tumors in patients with Lynch syndrome. Nowadays, MMR testing is compulsory because the presence of MSI has prognostic, predictive, and therapeutic implications, but represents only 5% of all mCRC and 10–15% of early-stage CRC [6,11]. Testing for MMR includes a multiplex polymerase chain reaction (PCR) assay, the “Bethesda Panel”, or a multiplex immunohistochemistry (IHC) assay, in order to demonstrate the absence of one of four MMR enzymes (MLH1, MSH2, MSH6, and PMS2). [15].

The deficiency of DNA MMR proteins results in an accumulation of mutations due to the absence of reparation of insertions and deletions. This is especially remarkable in the microsatellite regions of DNA, leading to the hypermutated phenotype and known microsatellite instability [16]. The tumor microenvironment (TME) of MSI CRC tumors includes a higher number of tumor-infiltrating lymphocytes (TILs), including T helper 1 (Th1) CD4+ and cytotoxic CD8+ T cells, as well as antigen-presenting cells (APCs). The number of T cells included is typically high and, in some cases, larger than the number of tumoral cells, and there is an upregulation of the expression of immune inhibitory ligands and receptors including CTLA-4, PD-1, PD-L1, LAG-3, and IDO [17]. The high mutational burden and immune infiltration produce a unique phenotype of MSI tumors with higher immunogenicity than MSS tumors, supporting the response to ICIs [18,19].

### 2.2. Current Treatments Approved

Several clinical trials have investigated the therapeutic role of PD-1 inhibitors in MSI mCRC. Table 1 summarizes FDA-approved clinical trials that have evaluated ICIs in MSI mCRC.

The phase II trial KEYNOTE-016 tested pembrolizumab, an antiPD-1, in three cohorts of patients: MSI mCRC, MSS mCRC, and MSI non-mCRC. The objective response rate (ORR) was 50% in MSI mCRC and 0% in MSS mCRC. The disease control rate (DCR) was 89% in the first group and decreased to 14% in MSS mCRC. The progression-free survival (PFS) and overall survival (OS) rates at 24 m were 61% and 66% in MSI mCRC, respectively [20]. Based on the results of this trial, in 2017 the Food and Drug Administration (FDA) and European Medicines Agency (EMA) approved pembrolizumab for patients with MSI mCRC after prior treatment with CT agents.

The phase III KEYNOTE-177 trial compared pembrolizumab to standard treatment of mCRC (5-fluorouracil-based CT with or without bevacizumab (anti-VEGF) or anti-EGFR) as the first line of treatment in MSI mCRC. After a median follow-up of 32.4 m, PFS was 16.5 m in patients treated with pembrolizumab compared with 8.2 m in patients receiving standard treatment (hazard ratio (HR):0.6; *p* = 0.0002). An ORR of 43.8% was observed in the pembrolizumab arm, higher than the ORR of 33.1% of the standard treatment arm. Amongst the patients who experienced response in the pembrolizumab group, 83% had an ongoing response at 24 m [21].

The final analysis of KEYNOTE-177 showed a median PFS (mPFS) of 16.5 m vs. 8.2 m (HR = 0.59) with an ORR of 45.1% vs. 33.1% between the pembrolizumab and CT groups, respectively. Although the OS was better in patients treated with pembrolizumab, the improvement did not reach statistical significance, probably due to the 60% crossover rate reported in the intention-to-treat (ITT) population [23].

Based on the results of the KEYNOTE-177 trial, in June 2020 and December 2020 FDA and EMA respectively approved pembrolizumab as first-line treatment for patients with MSI mCRC. According to the results of KEYNOTE-158 trial, pembrolizumab has an agnostic indication for the treatment of refractory patients with unresectable or metastatic solid tumors with high mutational burden (TMB-H) [≥10 mutations per megabase (mut/Mb)] [24].

The CheckMate 142 phase II trial evaluated nivolumab (antiPD-1) and/or ipilimumab (antiCTLA-4) in pre-treated and treatment-naïve MSI mCRC patients. This trial is also evaluating other combinations such as adding daratumumab, an anti-CD38, or relatlimab, an anti-LAG-3, to nivolumab, or combining ipilimumab/nivolumab with cobimetinib. Previously treated patients were assigned to receive either nivolumab as a single agent or a combination of nivolumab plus ipilimumab. The nivolumab arm had an ORR of 31.1%, with a DCR of 69%. The mPFS was 14.3 m, with a mOS not reached. The combination of ipilimumab/nivolumab in previously treated MSI mCRC patients showed an ORR of 65%, including 13% of complete responses (CR), and a DCR of 85%. The mPFS and OS were not reached, with 48 m PFS and OS rates of 53% and 71%, respectively [22].

CheckMate 142 also studied the combination of nivolumab plus ipilimumab as the first line of treatment in MSI mCRC, with a similar ORR to the pre-treated patients’ cohort (69%), including also 13% CR, and a DCR of 84%. The PFS and OS were not reached, with 24 m rates of 74% and 79%, respectively [25]. In July 2017, the FDA also approved nivolumab, either alone or in combination with ipilimumab, in second or further lines for patients with MSI mCRC, based on the results of these cohort studies. These alternatives have not been approved by the EMA.

The CheckMate 8 HW phase III trial (NCT04008030) compares nivolumab, nivolumab, and ipilimumab versus CT as treatment for MSI mCRC [26]. The primary endpoint is PFS for nivolumab plus ipilimumab vs. nivolumab across all lines, and nivolumab plus ipilimumab vs. CT in the first line of treatment.

The NRG-GI004/SWOG-S1610 COMMIT phase III trial (NCT02997228) investigated atezolizumab either alone or in combination with CT plus bevacizumab compared with CT plus bevacizumab as the first line of treatment for MSI mCRC [27]. However, the FOLFOX-bevacizumab arm was closed to enrolment due to emerging data [21]. The redesigned COMMIT trial was reactivated on January 2021 comparing either atezolizumab alone or in combination with FOLFOX-bevacizumab [28].

A phase II trial has investigated the activity of dostarlimab, an antiPD-1, in patients with MSI or POLE mutation in gastrointestinal tumors. The ORR of MSI patients was 38.7% (46% if considering only CRC), with 80.9% of responses ongoing after 18 m [29].

## 3. Current Landscape and Challenges of ICIs in MSS mCRC

### 3.1. Does ICIs Work in MSS mCRC?

About 95% of patients with mCRC present with pMMR and consequent MSS or MSI-L status (non-MSI-H). [30]. The use of ICIs in monotherapy or combination has obtained poor results in various clinical trials with MSS mCRC.

The inherent resistance to ICIs in the majority of MSS mCRC can be explained by a considerably lower TMB and an immune-desert TME with absent or inactive cytotoxic T lymphocytes and a low expression of checkpoint proteins (PD-1, PD-L1, CTLA-4, LAG-3) [31,32]. More than 10-fold lower mutation load with a subsequent low number of neoantigens is expressed compared to MSI [33]. This was evaluated in the KEYNOTE-016 study where genomic analysis demonstrated a mean of 1782 somatic mutations in MSI mCRC versus 73 mutations in MSS mCRC [34].

Another theory exposed is the immunoediting hypothesis by which better responses are achieved in early stage MSS CRC treated with ICIs. Cancer cells at these stages lose antigens to APCs and progressively avoid the immune system, so they proliferate and metastasize. There are other theories for explaining the higher absence of response when CRC is presented in the advanced stage: a higher degree of systemic immunosuppression is observed in metastatic tumors with downregulation of HLA molecules and a subsequent reduction of antigen presentation [35,36].

### 3.2. Challenges

For MSS mCRC, ICIs are being actively explored in combination with treatments that aim to increase intra-tumoral immune response and render the tumor “immune-reactive”. It is a priority to confirm the intrinsic mechanisms of resistance to immunotherapy in this subgroup, and furthermore necessary to identify relevant new biomarkers.

Effective immune response against tumor cells is a complex system, from the activation of the immune cells to their recruitment within the TME to execute their actions. Therefore, promoting the knowledge of the diverse mechanisms that avoid the activation of the immune system in response to ICIs in MSS mCRC is a keystone for construction novel combination therapies [30]. The scarce mutational load can be overcome by using different approaches: on the one hand, new therapies are emerging such as neoantigen vaccines or adoptive T-cell therapy which are specifically directed to the scarce tumor antigens expressed. On the other hand, the mutation burden can hypothetically be increased by different methods to make these tumors more sensitive to ICIs [37,38,39]. Other strategies aim to transform the immunosuppressive TME into an immune-responsive state with a working immune infiltrate, or to influence the interferon-γ (IFN-γ) signature through inhibiting the expression of immunosuppressive ligands [30].

We illustrate the challenges in the field of combination strategies with ICIs in MSS mCRC (Figure 1).

#### 3.2.1. Combination of Anti-VEGF Agents and Chemotherapy with ICIs

VEGF contributes to tumor angiogenesis and upregulates immune checkpoint molecules (PD-1, PD-L1, CTLA4 and LAG-3). It also downregulates antigen-presentation molecules and inhibits dendritic cell maturation [40,41]. Oxaliplatin in combination with an anti-VEGF drug enhanced the anti-tumor activity of PD-1 pathway blockade in mCRC [42]. Therefore, the combination of an anti-VEGF agent with ICIs may induce immune-stimulatory effects. The combination of CT, anti-VEGF, and ICIs may be effective due to the potential synergistic mechanisms [43,44]. The relevant clinical trials evaluating the use of anti-VEGF and CT plus ICIs are summarized in Table 2.

The MODUL phase II trial (NCT02291289) evaluated maintenance treatment with combined atezolizumab/bevacizumab/fluoropyrimidine after first-line induction with FOLFOX/bevacizumab in BRAF wild-type (BRAFwt) MSS mCRC. No differences were demonstrated in terms of PFS or OS compared with bevacizumab in combination with fluoropyrimidine alone [45].

The BACCI phase II trial (NCT02873195) assessed the effect in refractory mCRC of adding atezolizumab to capecitabine and bevacizumab. The majority of tumors (86.7% of mCRC in the atezolizumab and 85.7% in the control group) were MSS. A modest statistically significant improvement in PFS was detected (4.4 m vs. 3.3 m, *p* = 0.051) [46]. A phase II trial focusing only on MSS CRC patients used bevacizumab, capecitabine, and pembrolizumab (NCT03396926) [47]. The ORR in 40 evaluable patients was 5%, with an mPFS of 4.3 m and mOS of 9.6 m [47]. The CheckMate 9 × 8 phase II trial (NCT03414983) compared the addition of nivolumab to FOLFOX and bevacizumab as the first-line treatment in mCRC. ORR was 60% in the experimental group and 46% in the standard of care control with an mPFS of 11.9 m overall (*p* = 0.3) and a mOS of 29.2 m in patients treated with nivolumab and not reached in the standard of care group [48].

The AtezoTRIBE phase II study (NCT03721653) randomized mCRC patients to receive standard (FOLFOXIRI and bevacizumab) or experimental (FOLFOXIRI, bevacizumab and atezolizumab) first-line treatment, regardless of tumor microsatellite status [49]. A PFS benefit was attained with the addition of atezolizumab in the ITT population, but pMMR patients did not have a significative improvement in mPFS (12.9 m vs. 11.4 m, *p* = 0.072).

In conclusion, the combination of anti-VEGF, CT, and ICIs may be a useful strategy of treatment in MSS mCRC, but more clinical trials must demonstrate its effectiveness.

#### 3.2.2. Combination of Anti-EGFR Agents and Chemotherapy with ICIs

Cetuximab, a chimeric IgG1 antibody against EGFR, can evoke a T-cell-mediated anti-tumor immune response and stimulate NK-mediated antibody-dependent cellular cytotoxicity, independently of RAS mutational status [50]. However, panitumumab, the fully human IgG2 antibody anti-EGFR, has not demonstrated the same skills in mobilizing immune cells against tumor cells [51]. Table 3 summarizes clinical trials that have studied anti-EGFR, CT, and ICIs in MSS mCRC.

The AVETUX phase II trial (NCT03174405) studied avelumab, anantiPD-L1, cetuximab, and FOLFOX as the first line of treatment in RAS/BRAFwt mCRC patients, independently of microsatellite status, with 95% of patients being MSS. An 80% ORR was detected and the mPFS was 11.1 m [52].

The AVETUXIRI phase II trial (NCT03608046) evaluated avelumab, cetuximab, and irinotecan in refractory mCRC patients [53]. The DCR was 60.0% and 61.5% in RASwt and RASm, respectively, with an mPFS and mOS of 4.2 and 12.7 m in RASwt patients and 3.8 m and 14 m in RASm patients. These differences were statistically significant.

The CAVE colon phase II trial (NCT04561336) tested avelumab plus cetuximab as a rechallenge in previously treated RASwt mCRC patients. [54]. Patients were not selected for microsatellite status. The mOS and mPFS were 11.6 m and 3.6 m, respectively [55].

A phase II trial (NCT03442569) evaluated the combination of nivolumab, ipilimumab, and panitumumab in pretreated MSS RASwt mCRC. Patients were excluded if they had been treated with prior anti-EGFR therapy. The 12 w ORR was 35% and the mPFS was 5.7 m [56]. A phase I/II single-arm trial (NCT04017650) of nivolumab, cetuximab, and encorafenib in BRAF V600E mutant MSS mCRC patients who had progressed to at least one prior line of treatment showed an ORR of 45%, mPFS of 7.3 m, and mOS of 11.4 m [57].

The combination of anti-EGFR, avelumab, and triplet CT is under evaluation in the AVETRIC phase II study (NCT04513951). FOLFOXIRI plus cetuximab and avelumab is administered as first-line therapy in RASwt mCRC, followed by maintenance with 5-FU plus cetuximab and avelumab, regardless of microsatellite status [58].

Interpretation of the results of trials combining CT and targeted therapy is complicated because a great part of the outcomes could be due to the activity of the standard treatment, with the combination arms having shown scarce and non-statistically significant improvements.

**Table 3 cancers-15-00863-t003:** Currently ongoing and completed clinical trials of anti-EGFR, CT, and ICIs in MSS mCRC.

Study	Treatment	Phase	Endpoint 1	Setting	ORR (%)	mPFS (Months)	mOS (Months)	Status
AVETUX (NCT03174405) [52]	Avelumab + cetuximab + irinotecan	II	PFS	1st line mCRC	80%	11.1	NA	Completed
AVETUXIRI (NCT03608046) [53]	Avelumab + cetuximab + irinotecan	II	PFS	Refractory mCRC	60.0% (RASwt), 11.5% (RASm)	4.2 (RASwt), 3.8 (RASm)	12.7 (RASwt), 14 (RASm)	Completed
CAVE Colon (NCT04561336) [55]	Avelumab + cetuximab	II	OS	Refractory RASwt mCRC	NA	3.6	11.6	Completed
NCT04017650 [57]	Nivolumab + cetuximab + encorafenib	I/II	ORR	BRAF V600E mutant pMMR mCRC	45%	7.3	11.4	Active, not recruiting
AVETRIC (NCT04513951) [58]	FOLFOXIRI + cetuximab + avelumab	II	PFS	1st line mCRC	NA	NA	NA	Recruiting

Abbreviations: ORR: objective response rate; OS: overall survival; PFS: progression-free survival; BOR: best objective response; NR: not reached; NA: not available.

#### 3.2.3. Combination of Temozolomide with ICIs

Temozolomide (TMZ) is an oral alkylating drug that acts by methylating DNA strands at the O6 position of guanine, which damages the DNA and inhibits its replication. Its activity is linked with the O6-methylguanine methyltransferase (MGMT) enzyme, which reduces the therapeutic efficacy of TMZ [59]. The epigenetic silencing of MGMT, mediated by the methylation of its promoter region, is involved in low DNA repair of O6-alkylguanine adducts, thereby improving the sensitivity of cancer cells to TMZ [60].

MGMT methylation is detected in 30–40% of CRC and is strongly associated with RAS mutations [61]. Several phase II trials have demonstrated that TMZ is an active option in pretreated mCRC patients [59,62,63,64]. TMZ is not yet a standard treatment, because diverse drug schedules were used in the trials and the definition of MGMT methylation remains unsettled [65,66].

TMZ can induce somatic mutations in MMR genes in solid tumors [67,68] and produce depletion of T-regulator lymphocytes and activation of cytotoxic T lymphocytes [69]. A hypermutated phenotype and a high load of neoantigens were observed in mCRC cells initially responsive to TMZ at the time of acquired resistance [70]. The rationale to explore combining TMZ and ICIs in MSS mCRC patients is the activation of effective immune surveillance by TMZ, so TMZ would change a cold tumor into a hot one.

Clinical trials evaluating combinations of TMZ and ICIs are summarized in Table 4. The MAYA phase II trial (NCT03832621) studied the combination of nivolumab, ipilimumab, and TMZ in patients with MSS MGMT-silenced mCRC who had not progressed after two cycles of TMZ, independently of RAS mutational status. Among 716 pre-screened patients, 204 (29%) were molecularly eligible and 135 of them started the first part of the treatment. Among these, 102 (76%) had to discontinue because of death or disease progression in the TMZ priming phase, whereas 33 patients (24%) achieved disease control and started the second part of the treatment. These patients represented the final study population with mPFS and mOS of 7.0 m and 18.4 m, respectively, and an ORR of 45% [71].

The ARETHUSA phase II trial (NCT03519412) is a non-randomized study with two cohorts in which patients with refractory MSI mCRC are treated with pembrolizumab until progression, and patients with MSS, RASm and MGMT-silenced mCRC receive TMZ until progression [72]. A biopsy is performed when disease progression is confirmed in the MSS cohort in order to determine TMB, and patients with TMB > 20 mut/Mb receive pembrolizumab. The primary endpoint is ORR in the MSS cohort treated with pembrolizumab, with the MSI cohort used for indirect comparison. The estimated study completion date is December 2023.

Another phase II clinical trial (NCT04457284) is evaluating the combination of TMZ, cisplatin, and nivolumab in MSS mCRC refractory to prior lines of treatment [73].

#### 3.2.4. Combination of ICIs with DNA Damage Response (DDR) Inhibitors

The DNA damage response (DDR) is a complex mechanism that recognizes an error within the genome and triggers several pathways to activate cell-cycle checkpoints in order to repair the DNA before continuing with the division. When the DNA is irreparably damaged, DDR directs these cells to apoptosis or permanent anergy status [74]. Poly-ADP-ribose polymerase 1 (PARP1) mRNA overexpression was observed in 70.3% of a series of CRC analyzed, most at early stages. A hypothesis of the role of PARP1 in CRC oncogenesis has been postulated from these results [75]

By inhibiting PARP with inhibitors (PARPi) the single-strand DNA cannot be repaired, and consequently an accumulation of DNA damage generates release of neoantigen load which increases TMB and PD-L1 expression, with a potential synergistic effect in combination with ICIs [76]. There have been promising results with the use of PARPi with ICIs in various solid tumors, including MSS mCRC (Table 5). The DAPPER phase II basket trial evaluates the combination of durvalumab with olaparib (PARPi) or cediranib (VEGFR inhibitor) in refractory MSS mCRC, pancreatic carcinoma, or leiomyosarcoma. The DAPPER trial examines the changes in the genomic and immune biomarkers in the baseline biopsy and first treatment biopsy, as well as in peripheral blood and stool samples [77].

Another phase I/II basket study is testing the same drugs as DAPPER and also the combination of the three (NCT02484404). The primary endpoints are ORR and determining the tolerability of the combinations [77]. In ovarian cancer, the preliminary results show a DCR of 67% with 44% partial responses and acceptable tolerability [78]. Other phase I and II trials are also evaluating the combination of PARPi with ICIs in refractory patients with mutations in the homologous recombination repair genes (NCT04123366, NCT03842228) and patients without them (NCT03772561) [79,80,81].

Other molecules involved in DDR include ATR, which also acts on single-strand damage recognition and prevents the cell cycle from progressing to the G2 phase from the S phase by activating the checkpoint kinase 2. A phase I/II trial (NCT04266912) is evaluating the safety of combining an ATR inhibitor berzosertib with avelumab in solid tumors with actionable aberrations in one or more of the DNA DDR genes, including ARID1A, ATM, or ATR, among others. [82]. Another potential target is WEE1, which is involved in preventing the transition from G2 to M by negatively regulating the checkpoint kinase 1. A WEE1 inhibitor, adavosertib, is being tested with durvalumab in a phase I trial, with results pending [82].

**Table 5 cancers-15-00863-t005:** Currently ongoing and completed clinical trials of DDR inhibitors and ICIs in MSS mCRC.

Study	Treatment	Phase	Endpoint 1	Setting	ORR (%)	mPFS (Months)	mOS (Months)	Status
DAPPER (NCT03851614) [77]	Durvalumab + Olaparib/Cediranib	II	Changes in genomic and immune biomarkers	A refractory solid tumor (only pMMR CCR)	NA	NA	NA	Active, not recruiting
NCT02484404 [77]	Durvalumab + Olaparib +/o Cediranib	I/II	ORR Safety and tolerability, MTD	Refractory solid tumors	NA	NA	NA	Recruiting
NCT04123366 [81]	Olaparib + Pembrolizumab	II	ORR	Refractory solid tumor + mutation in HRR/HRD	NA	NA	NA	Recruiting
NCT03842228 [80]	Copanlisib + Olaparib + durvalumab	I	MTD	Refractory solid tumor + germline or somatic mutations in DDR genes	NA	NA	NA	Recruiting
NCT03772561 [79]	AZD5363 + Olaparib + Durvalumab	I	ORR/BOR	Refractory solid tumors	NA	NA	NA	Recruiting
NCT04266912 [83]	Avelumab and Berzosertib	I/II	Safety and tolerability, MTD	Refractory solid tumors with a mutation in DDR genes	NA	NA	NA	Recruiting
NCT02617277 [82]	Adavosertib + Durvalumab	I	DLTs	Refractory solid tumors	NA	NA	NA	Active, not recruiting

Abbreviations: DLT: dose-limiting toxicity; MTD: maximum tolerable dose; DCR: disease control rate; IDCR: immune disease control rate; RD: recommended dose; ORR: objective response rate; OS: overall survival; PFS: progression-free survival; nr: not reached; NA: not available.

#### 3.2.5. Combination of Multikinase Inhibitors (Anti-VEGFR) with ICIs

The combination of ICIs with vascular endothelial growth factor receptors (VEGFR) inhibitors may represent a useful approach due to the immunosuppressive role that VEGF has in the TME. VEGF promotes angiogenesis within the tumor with an expansion of tumor-suppressing immune cells and tumor-associated macrophages (TAMs) that supports the characteristic immune evasion. Furthermore, it decreases T cell activity by promoting the expression of the immune checkpoints PD-1, CTLA-4, TIM3, and LAG3 on their surface. Therefore, there is a demonstrable basis for combining inhibitors of VEGF/VEGFR with ICIs [84,85].

Regorafenib is a multikinase inhibitor (MKI) against VEGFR1, VEGFR2, VEGFR3, epidermal growth factor homology domain 2 (TIE-2), platelet-derived growth factor receptor-β (PDGFR β), c-kit, RET, RAF-1, colony-stimulating factor-1 (CSF-1R), and BRAFwt and V600E mutation. [86,87] Its potential synergistic role with ICIs lies in its immunomodulatory properties through the inhibition of CSF1R, a tyrosine kinase receptor that is involved in TAM proliferation and differentiation. These cells may be presented as M1 macrophages that promote antitumor actions, or M2 macrophages which are involved in immune evasion favoring oncogenesis. The latter are downregulated with the inhibition of CSFR1 in the presence of regorafenib: M2 TAMs express CD206 on their membrane and have been shown to be considerably reduced in CRC murine models, with a significant increase of M1 TAMs [87,88]. The effects on tumor reduction as well as on reducing intratumoral macrophages are synergistically enhanced with antiPD-1 treatment, as shown in preclinical models. Additionally, a reduction of regulatory T cells was observed with the addition of regorafenib to antiPD-1 treatment, promoting an antitumor effect [88,89]. Interestingly, the benefits obtained from regorafenib are maintained by the presence of antiPD-1 treatment when the treatment is discontinued [89]. Table 6 describes the clinical trials that have studied the activity of MKI plus ICIs in MSS mCRC.

The REGONIVO phase Ib trial combining regorafenib and nivolumab included patients with metastatic gastric and mCRC on progression to two or more lines of treatment. The primary endpoint (ORR) was encouraging (33%) in the subset of MSS mCRC patients (*n* = 24) [90]. Nonetheless, REGNIVO, the phase II study from America, failed to produce similar results (ORR = 7%) [91]. Other clinical trials have also published poor results: in REGOMUNE, a single-arm phase II study testing regorafenib in combination with avelumab in pretreated MSS mCRC patients, no objective response was achieved, with stable disease as the best outcome (54% of patients). However, considering the results of regorafenib as a single agent, the addition of the antiPD-1 improved the PFS and OS results (median PFS and OS of 3.6 and 10.8 m, respectively, vs. 1.9 m and 6.4 m observed in the CORRECT trial) [92,93]. Another phase I/Ib study also tested regorafenib and nivolumab in refractory MSS mCRC patients, with only 10% of patients achieving partial response, but >50% of patients had stable disease as the best response, with considerable control of the disease [94]. The combination with pembrolizumab in a phase I/II clinical trial on refractory patients also showed no objective response, with stable disease in 49% of the patients. Although it did not meet its primary endpoint (PFS), the OS was 10.9 m, similar to the value observed in the REGOMUNE trial. [95] Further studies are required to establish the role of regorafenib in this subgroup of patients.

Lenvatinib is an MKI that targets VEGFR 1–3, RET, FGFR 1–4, c-KIT, and PDGFRα. Due to its immunomodulatory activity, it has been assessed in combination with pembrolizumab in other malignancies including renal cell carcinoma, with remarkable results which positioned it as part of the standard of care in first-line treatment [96]. In the single-arm phase II study LEAP-005, MSS mCRC patients on progression to standard lines of therapy received lenvatinib in combination with pembrolizumab, and a substantial ORR of 22% was reported. [97] The phase III trial (LEAP-017) comparing lenvatinib–pembrolizumab with the standard of care regorafenib or TAS-102 is currently ongoing, with pending results [98]. As with regorafenib, the antitumor activity of lenvatinib and pembrolizumab is enhanced by the combination of both drugs, with an increase in the percentage of CD8+ T cells and a reduction of M2 TAMs demonstrated in preclinical models. It is hypothesized that the dual inhibition of VEGF and FGFR is involved not only in inhibiting tumor angiogenesis but also in converting the immunosuppressive TME to a more immunogenic status by increasing IFN-γ production by CD8+ T cells [99].

The use of other MKIs that target VEGFR, such as cabozantinib, has not demonstrated improvement on previous results but reinforces the idea of the synergy between antiVEGFR and ICI; the phase II CAMILLA trial assessed the combination of cabozantinib with durvalumab in MSS mCRC patients who had progressed from two or more prior therapies, with a considerable ORR of 27.6% and similar rates of PFS and OS (3.8 m and 9.1 m, respectively) [100,100]. The phase Ib COSMIC-021 study evaluated the efficacy of combining cabozantinib and atezolizumab, an antiPD-L1, in pretreated solid tumors; the ORR was 9.7% in MSS patients, considerably lower than that reported in the CAMILLA trial. However, in the subgroup analysis, the patients with RASwt tumors had an ORR of 25%, coinciding with the higher ORR presented in this subgroup in the CAMILLA trial (ORR of 50%) [100,101], which may indicate a higher response rate in subgroups of patients within MSS mCRC; a retrospective study of patients with MSI CRC tumors revealed a less immunogenic TME in RASm [102]. Nonetheless, the results of the combinations were much better than those observed with MKI in monotherapy, suggesting a synergistic mechanism with ICIs.

Fruquintinib is a highly selective oral inhibitor of VEGFR 1–3 that recently proved its benefits on OS in pre-treated mCRC. A retrospective study comparing the combination of fruquintinib with antiPD-1 vs. regorafenib with antiPD-1 agents, including toripalimab, nivolumab, sintilimab, or camrelizumab, was performed by Sun et al., reporting better results in the first group in terms of PFS (6.4 m vs. 3.9 m), although the ORR was slightly better in the second group (7.1% vs. 8.7%). Curiously, as described previously, the presence of RAS mutations, as well as other factors such as the presence of liver metastases or a right colon localization, were associated with worse responses, although no significant differences in PFS were observed between the subgroup of patients with and without these characteristics [103]. Other studies have not obtained the same results comparing regorafenib and fruquintinib in combination with an antiPD1, although favorable results make them both a possible approach in refractory MSS mCRC, with studies currently ongoing (NCT04577963 and NCT04483219) [104,105,106].

**Table 6 cancers-15-00863-t006:** Currently ongoing and completed clinical trials of multikinase inhibitors (Anti-VEGFR) and ICIs in MSS mCRC.

Study	Treatment	Phase	Endpoint 1	Setting	ORR (%)	mPFS (Months)	mOS (Months)	Status
REGONIVO (NCT03406871) [90]	Nivolumab + Regorafenib MSS mCRC cohort	I/II	MTD and RD	Refractory to standard treatments (≥3rd line)	33.3% (MSS patients)	7.9	NR	Completed
REGNIVO (NCT04126733) [91]	Nivolumab + Regorafenib	II	ORR	Refractory to standard treatment	7.1%	2	13	Completed
REGOMUNE (NCT03475953) [92]	Avelumab+Regorafenib MSS mCRC cohort	I/II	I: Safety and tolerability II: DCR	Refractory to standard treatment	NA	3.6	10.8	Recruiting
NCT03712943 [94]	Nivolumab + Regorafenib	I	DLT and MTD	Refractory to standard treatment	10%	4.3	11.1	Active, not recruiting
NCT03657641 [95]	Pembrolizumab + Regorafenib	I/II	DLT, PFS, and OS	Refractory to standard treatment	0%	2	10.9	Active, not recruiting
LEAP-005 (NCT03797326) [97]	Pembrolizumab + Lenvatinib MSS mCRC cohort	II	ORR	Refractory to standard treatments (≥3rd line)	22%	2.3	7.5	Active, not recruiting
LEAP-017 (NCT04776148) [98]	Pembrolizumab + Lenvatinib vs. TAS-102/Regorafenib	III	OS	Refractory to standard treatment	NA	NA	NA	Active, not recruiting
CAMILLA (NCT03539822) [100]	Cabozantinib + Durvalumab mCRC cohort (all MSS)	I/II	MTD and ORR	Refractory to standard treatments (≥3rd line)	27.6%	4.4	9.1	Recruiting
COSMIC-O21 (NCT03170960) [101]	Cabozantinib + Atezolizumab	I/II	MTD and ORR	Refractory to standard treatment	9.7%	3	14	Active, not recruiting
NCT04483219 [106]	TKI (fruquintinib or regorafenib) + Toripalimab	II	9-month PFS	Refractory to standard treatment	NA	NA	NA	Recruiting
NCT04577963 [105]	Fruquintinib+ tislelizumab	I/II	ORR	IO-Naïve	NA	NA	NA	Recruiting

Abbreviations: DLT: dose-limiting toxicity; MTD: maximum tolerable dose; DCR: disease control rate; iDCR: immune disease control rate; RD: Recommended dose; ORR: objective response rate; OS: overall survival; PFS: progression-free survival; BOR: best objective response; NR: not reached; NA: not available.

#### 3.2.6. Targeting MAPK Pathway in Combination with ICIs

It is known that the mitogen-activated protein kinase (MAPK) pathway, composed of the RAS/RAF/MEK/ERK signaling cascade, plays an essential role in cellular proliferation and survival. RAS is one of the most frequently mutated oncogenes in numerous neoplasms including CRC [107]. One of the hypotheses put forward is that mutations on the MAPK axis result in the upregulation of expression of immune suppressive checkpoints such as PD-L1 along with downregulation of class I MHC on the tumor surface, with a correlated decrease of intratumoral T-cell infiltration [108,109,110]. Therefore, this rationale is based on the inhibition of that axis in combination with the effects of ICIs. The main clinical trials are summarized in Table 7.

The role of MEK inhibitors in combination with ICIs has been researched in preclinical studies, with an observed increase of the CD8+ T-cell population and class I MHC expression, resulting in better responses in comparison with MEK inhibitors in monotherapy [110,111]. Hence, the phase III IMblaze 370 trial, which involved up to only 5% MSI mCRC cases, assessed the combination of encorafenib, a MEK1 and MEK2 inhibitor, plus atezolizumab vs. atezolizumab vs. regorafenib in a third or subsequent line of treatment. The primary endpoint was OS, but no benefit was observed in the combination group in comparison with regorafenib (8.87 m vs. 8.51 m) or for atezolizumab and regorafenib (7.1 m vs. 8.51 m), nor were there significant differences in PFS or ORR [112]. Possible explanations for these results were proposed: on the one hand, the outcomes from the regorafenib group were better than those from the CORRECT trial so the possible benefit may have been masked. On the other hand, MEK inhibitors may not be effective enough to achieve the necessary immunomodulation to enhance the ICIs’ effect, as immunosuppressive actions associated with MEK inhibitors such as inhibiting T-cell proliferation have been described in other preclinical studies. Meanwhile, MAPK gene expression may be involved: in the phase I study of atezolizumab plus cobimetinib, cases with MAPK gene expression >50% had better mPFS and mOS in comparison with lower expressions, which could represent a biomarker of response, although further research is required [113,114].

Preliminary results from a phase II study combining pembrolizumab, binimetinib, a MEK1 and MEK2 inhibitor, and bevacizumab in third or subsequent treatment lines reported scarce activity with an ORR of 13% [115]. A phase I dose-escalation study in refractory solid tumors testing selumetinib, another MEK1 and 2 inhibitor, plus durvalumab has advanced to a phase II where a cohort of MSS mCRC has been enrolled [116,117].

The inhibition of other steps of the cascade signaling has also been investigated in the MSS setting. Approximately 60% of the somatic inactivation of MMR genes is co-presented with a BRAF mutation (BRAFm). The tumorigenesis of BRAFm tumors is related to extensive DNA methylation of the CpG islands and MLH1 promoter methylation that causes an MSI phenotype [118,119]. Only 5–10% of MSS tumors co-occur with a BRAF mutation and the addition of ICIs to the targeted therapy is based on the increase of T cell infiltration caused by inhibiting BRAF with BRAF, MEK, and EGFR inhibitors. The potential synergistic action of the combination has been tested in vivo with a benefit in response compared with their separate use. A phase I trial is testing the combination of dabrafenib and trametinib plus spartalizumab, an antiPD-1, in BRAFV600E mCRC patients. An ORR of 42 has been reported, with one complete response notified. Biopsies at baseline and day 15 were analyzed, with an increase in T-cell infiltration and expression of cytotoxic genes observed [120,121]. Another phase I study is assessing the combination of dabrafenib with LTT462, an ERK inhibitor, plus spartalizumab/tislelizumab, both antiPD-1, with results pending [122].

Approximately 45% of CRC has a KRAS mutation (KRASm), leading to constant MAPK activation and T-cell exclusion. Recently, sotorasib, a KRAS inhibitor against KRAS p.G12C, has emerged with promising results in pretreated non-small cell lung cancer, with an ORR of 37.1% and DCR of 80.6%. However, the outcomes were poorer in pretreated mCRC, with an ORR of 9.7%; this inconsistency may be explained by the presence of other oncogenic drivers in CRC, and combination strategies should be explored [123,124,125]. As mentioned earlier, KRAS mutations are linked to upregulation of PD-L1 and combination treatment with antiPD-1/L1 is being explored in the CodeBreaK100 trial [126]. A phase I/II study (KontRASt-01) is currently evaluating the combination of tislelizumab, an antiPD-1, with JDQ443, another KRAS G12C inhibitor, with or without TNO155, an SHP2 inhibitor [127]. SHP2 is a tyrosine phosphatase that plays a key role in the carcinogenesis of KRASm CRC, with an antitumor effect demonstrated both in vitro and in vivo [128].

#### 3.2.7. Targeting PIK3CA/AKT/mTOR Pathway

Derived either from mutations in *PIK3CA* (PIK3CAm) or from the loss of PTEN, PI3K pathway activation can be found in approximately 20% of CRC. Preclinical studies have shown that tumoral cells with activations on this pathway are associated with increased immune evasion and resistance to immunotherapy due to an upregulation of CD4+ T cells as well as reprogramming of TAMs to their immunotolerant subtype (M2) [129]. Consequently, to cope with this mechanism of primary resistance, a combination of an inhibitor of PI3K (copanlisib) with nivolumab has been proposed in a phase I/II study in pretreated patients, with a second phase in which the primary endpoint is a 6-month ORR in MSS CRC patients, either PIK3CA wild-type or PIK3CAm [130,131].

**Table 7 cancers-15-00863-t007:** Currently ongoing and completed clinical trials of inhibitors of MAPK/PIK3CA and ICIs in MSS mCRC.

Study	Treatment	Phase	Endpoint 1	Setting	ORR (%)	mPFS (Months)	mOS (Months)	Status
NCT01988896 [114]	Atezolizumab + Cobimetinib	I	Safety and tolerability, MTD	Refractory solid tumors	8%	1.9	9.8	Completed
IMblaze 370 (NCT02788279) [112]	Atezolizumab + Cobimetinib vs. Atezolizumab vs. Regorafenib	III	OS	Refractory to standard treatments (≥3rd line)	3%	1.9 vs. 1.9 vs. 2	8.9 vs. 7.1 vs. 8.5	Completed
NCT02586987 [117]	Selumetinib + Durvalumab +/- Tremelimumab	I	DLT Safety and tolerability	Refractory solid tumors	NA	NA	NA	Completed
NCT03668431 [120] MSS patients	Dabrafenib + Trametinib + PDR001 (Spartalizumab)	II	ORR	First or subsequent lines BRAFV600E	42%	NA	NA	Recruiting
NCT04294160 [122]	Dabrafenib + LTT462 (ERK inhibitor) + PDR001 (Spartalizumab)/Tislelizumab [122]	I	DLT Safety and tolerability	≥Second line BRAFV600	NA	NA	NA	Recruiting
CodeBreaK 100 (NCT03600883) [126]	Sotorasib + AntiPD-1/L1	I/II	DLT Safety and tolerability ORR	Refractory KRAS G12C	NA	NA	NA	Active, not recruiting
KontRASt-01 (NCT04699188) [127]	Tislelizumab + JDQ443 +/- TNO155	I/II	DLT Safety and tolerability	Refractory to standard treatments	NA	NA	NA	Recruiting
NCT03711058 [130]	Copanlisib + Nivolumab	I/II	MTD ORR	Refractory to standard treatments (≥third line)	NA	NA	NA	Active, not recruiting

Abbreviations: DLT: dose-limiting toxicity; MTD: maximum tolerable dose; DCR: disease control rate; iDCR: immune disease control rate; RD: recommended dose; ORR: objective response rate; OS: overall survival; PFS: progression-free survival; NR: not reached; NA: not available.

#### 3.2.8. Combination with Other ICIs 

As previously mentioned, monotherapy with antiPD-1 failed to prove efficacy in MSS tumors [34,132]. Therefore, the dual blockade with antiCTLA4 was tested due to the better results obtained in other neoplasms such as melanoma or renal cell carcinoma, attributed to the CTLA4 expression on regulatory T-cell surfaces [133]. A phase II trial assessed the combination of durvalumab and tremelimumab vs. BSC in both MSS and MSI mCRC refractory to all standard treatments. A disappointing global ORR of 0.8% was observed, although the OS had a trend in favor of the combination (6.6 m vs. 4.1 m). In the subgroup analysis, MSS patients had significantly greater OS and those who possessed high TMB (defined as ≥28 Mut/Mb), accounting for 21% of the patients, experienced the best results in terms of OS (HR: 0.34; *p* = 0.004) [134]. Following these findings, further clinical trials have tried to demonstrate the role of TMB in responding to ICI: the phase II basket study TAPUR analyzed the effect of nivolumab plus ipilimumab in pretreated mCRC with high TMB, defined as ≥9 Mut/Mb: DCR and ORR were both 10% and the trial closed due to futility [25,135].

An expanded phase I clinical trial evaluated the combination of botensilimab (next-generation antiCTLA-4) and balstilimab (antiPD-1) in pretreated MSS mCRC. A total of 41 patients were evaluated with an ORR of 24%, higher (42%) in patients with no liver metastases or locally-treated with resection or ablation techniques [136,137]. These results contribute to the theory that the presence of liver metastases does influence the response to immunotherapy; these lesions are thought to diminish tumor-specific CD8+ T-cell recount, lessening the efficacy of immunotherapy. This has been studied in preclinical models, and CD11b+F4/80+ macrophages presented in the liver are believed to be the main factor responsible for CD8+ T-cell apoptosis by inducing the Fas–FasL pathway. Hence, these liver macrophages induce systemic immunosuppression, and the hypothesis is postulated that treating these lesions with local therapies may help increase the response to immunotherapy [138].

Combinations with other ICIs have also been tested, also with poor results. The association of favezelimab, an antiLAG-3 antibody, with pembrolizumab in a phase I study in pre-treated mCRC reported four partial responses and one complete response (ORR 6.3%). Curiously, patients with CPS ≥1 presented a mOS of 12.7 m, almost twice that of patients with CPS < 1 (6.7 m) [139]. The main clinical trials are reviewed in Table 8.

#### 3.2.9. Combination of ICIs with Radiotherapy

As radiotherapy is predominantly a local therapy, its combination with other systemic therapies such as CT or immunotherapy aims to benefit from their systemic effects following radiation of a localized area. The destruction of tumoral cells liberates neoantigens and activates tumor-infiltrating dendritic cells, which triggers STING-mediated type I interferon production. A local immune response is activated, with a subsequent activation elsewhere, aimed at achieving systemic control by the so-called abscopal effect. The addition of immunotherapy can help trigger and increase the abscopal effect by blocking the checkpoint inhibitors [85]. This has been demonstrated in murine models of melanoma and non-small-cell lung cancer, with a benefit obtained from the use of antiCTLA4 in monotherapy, which is usually the most effective treatment for these kinds of tumors [140]. A case of tumor regression of non-irradiated distant lesions with the use of pembrolizumab and radiotherapy was described in a non-randomized phase II study involving pre-treated patients (Table 9). However, that was the only response achieved by the abscopal effect with this combination (ORR 9%) [141]. Similarly, the combination of ipilimumab and nivolumab with radiotherapy during the second cycle presented an ORR of 7.5% and a DCR of 17.5%. [142]. Another phase II study assessed the combination of durvalumab and tremelimumab with concurrent radiotherapy. Even though abscopal responses were observed (ORR = 8.3%), the study did not meet the prespecified endpoint criteria necessary to carry out a phase III follow-up [143].

The lack of effective response may be explained by the fact that the TME of MSS mCRC is characteristically cold, and may require further strategies between radiotherapy and ICI, such as the use of vaccines including the most immunodominant antigen in order to make the TME more immunogenic and hence overcome the intrinsic resistance to ICIs expressed by MSS mCRC [144]. An ongoing clinical trial aims to demonstrate the efficacy of using hypofractionated stereotactic ablative radiotherapy, as this seems to increase the effectiveness of ICIs [145].

Where the use of radiotherapy or ICIs in monotherapy would not obtain any response, a synergic effect is presented. Identifying additional biomarkers to identify the patients that could benefit from these strategies could be useful for the development of new clinical trials. As mentioned earlier, the presence of liver metastases is thought to lessen the response to immunotherapy, due to the specific microenvironment that collects and inactivates CD8+ T cells. Hence, the use of radiotherapy on liver metastases may overcome this acquired resistance mechanism and induce immune sensitivity [138].

**Table 9 cancers-15-00863-t009:** Currently ongoing and completed clinical trials of RT and ICIs in MSS mCRC.

Study	Treatment	Phase	Endpoint 1	Setting	ORR (%)	mPFS (Months)	mOS (Months)	Status
NCT03122509 [143]	Radiotherapy + Durvalumab + Tremelimumab	II	ORR	Refractory to standard treatments (≥3rd line)	8.3%	1.8	11.4	Completed
NCT03104439 [142]	Radiotherapy + Nivolumab + Ipilimumab	II	DCR (17.5%)	Refractory to standard treatments (≥3rd line)	7.5%	NA	NA	Recruiting
NCT02992912 [145]	Atezolizumab With Stereotactic Ablative Radiotherapy	II	PFS	Refractory	NA	NA	NA	Unknown

Abbreviations: DCR: disease control rate; iDCR: immune disease control rate; ORR: objective response rate; OS: overall survival; PFS: progression-free survival; NA: not available.

#### 3.2.10. Blocking TGF-β and Wnt Pathway + ICIs

Mutations in several signaling pathways have been associated with CRC proliferation and invasion, including transforming growth factor-beta (TGF-β), wingless-type MMTV integration site family (Wnt), EGFR, and p53. [146]

Genomic alterations in the TGF-β signaling pathway can be found in up to 27% of the non-hypermutated CRC. [14] TGF-β is a cytokine secreted by CRC cells and as it occurs with VEGF acts as a promoter of the tumorigenesis through multiple mechanisms, including promoting angiogenesis and immune evasion within the TME through direct and indirect actions on T cells, dendritic cells, and/or through regulating certain cytokines and extracellular matrix proteins. It impacts on T-cell differentiation towards a Th1 effector phenotype by repressing T-bet, the essential transcription factor to achieve this phenotype, and inhibits CD8+ T-cell responses. Therefore, by blocking TGF-β we can confer a more immunogenic TME that may allow ICIs to function. In preclinical models with mutations in three or the four of the pathways described, prominent T-cell exclusion and elevated TGF-β activity were demonstrated and the use of Galunisertib, a potent and selective TGF-β receptor I kinase inhibitor, reduced tumor size and prevented the appearance of liver metastases [146]. Vactosertib is another TGF-β receptor I kinase inhibitor that is being tested along with pembrolizumab in MSS mCRC refractory to all treatments. An ORR of 15.2% including five partial responses has been described, with the median duration of response not reached among the patients [147]. However, the use of bispecific antibody Bintrafusp-α (TGFβ-trap and anti-PD-L1) in MSS liver-limited mCRC treated with surgery and standard perioperative therapy failed to demonstrate its primary endpoint (ctDNA clearance), with associated large-volume recurrence [148]. 

Another hypothesis supported by the results from a study on squamous cell carcinoma tumor cell lines is that antiPD-1 therapy induces the known cytotoxic T-cell activity and also a competitive TGFβ-driven immunosuppressive program, and therefore there is a benefit to be gained from using a TGF-β inhibitor [149]. Following these findings, a phase I/Ib study is evaluating the use of NIS793, a fully human monoclonal antibody that binds with high affinity to TGFβ1 and TGFβ2, and with lower affinity to TGFβ3, as a single agent and in combination with spartalizumab, an antiPD-1 [150].

Other TGF-β inhibitors and monoclonal antibodies that bind to TGFβ are currently under investigation, with results pending (NCT03192345, NCT02423343, NCT03821935). Clinical trials underway or completed are listed in Table 10.

The activation of the Wnt pathway occurs in over 90% of MSS CRC patients and is associated with lower recruitment of T-cells. It may be caused by mutations in β-catenin or by adenomatous polyposis coli (APC) or R-spondin (RSPO) gene fusions, which are mutually exclusive [151]. As with KRAS mutations, Wnt pathway activation has been shown to confer T-cell exclusion, and a combination of Wnt-signalling inhibitors or small-molecule β-catenin inhibitors with ICIs may therefore have therapeutical potential. E7386 is a selective inhibitor of the interaction between β-catenin and CREB binding protein and is essential in the Wnt/β-catenin signaling pathway, with synergistic activity with antiPD-1 demonstrated in cell lines. This combination is being tested in refractory MSS CRC in a phase I/II study, with results pending (NCT05091346) [152,153].

**Table 10 cancers-15-00863-t010:** Clinical trials including blocking TGF-β, Wnt pathway plus ICIs in MSS mCRC.

Study	Treatment	Phase	Endpoint 1	Setting	ORR (%)	mPFS (Months)	mOS (Months)	Status
NCT03436563 [154]—cOHORT D	Anti-PD-L1/TGFbetaRII Fusion Protein M7824 (Bintrafusp-α)	Ib/II	Clearance of ctDNA	Completion of standard-of-care perioperative therapy	0%	NA	NA	Completed (other cohorts active, not recruiting)
NCT03724851 [147]	TEW-7197 (Vactosertib) + Pembrolizumab	I/II	MTD	Refractory to standard treatment	15.2%	NA	NA	Active, not recruiting
NCT02947165 [150]—Group 4	NIS793 +/- PDR001 (spartalizumab)	I/Ib	Incidence of DLTs and safety and tolerability	Refractory to standard treatment	NA	NA	NA	Completed
NCT03192345 [155]	SAR439459 + Cemiplimab	I	DLT and ORR	Refractory to standard treatment	NA	NA	NA	Terminated (competitive landscape and toxicity)
NCT02423343 [156]	Galunisertib +/- Nivolumab	I/II	MTD	Refractory solid tumors	NA	NA	NA	Completed

Abbreviations: DLT: dose-limiting toxicity; MTD: maximum tolerable dose; DCR: disease control rate; iDCR: immune disease control rate; RD: recommended dose; ORR: objective response rate; OS: overall survival; PFS: progression-free survival; NR: not reached; NA: not available.

#### 3.2.11. Other Combinations: Epigenetic Therapy and Novel Agents

Other strategies are being developed trying to turn these tumors into hot tumors (Table 11): the hypermethylation of DNA silences genes associated with the expression of cancer testis antigens and endogenous retroviruses, which enable T-cell and B-cell immunoreactivity by stimulating a state of viral mimicry. When not silenced, this results in an innate immune response that produces type I and type III interferons and other cytokines that attract CD8+ T cells to the TME [157]. Hence, a DNA methyltransferase inhibitor (DNAMi) can re-express these genes and make the tumor more sensitive to immunotherapy. Azacitidine is a DNAMi that has been tested with pembrolizumab in chemotherapy-refractory MSS Mcrc in a phase II trial, with ORR as the primary endpoint: a modest ORR of 3% was achieved, with an Mpfs of 1.9 m [158]. Another mechanism to prevent silencing these genes involves histone-deacetylase inhibitors (HDAi), enzymes which remove the acetyl groups on histones and reduce gene expression [157]. A combination of zabadinostat and nivolumab is being tested in refractory MSS Mcrc in a phase II trial; Mos of 7 m was reported with a DCR of 48% in the 3-year data [159,160]. Another HDAi, entinostat, has been assessed with pembrolizumab in a phase II study with ORR as the primary endpoint: only one partial response has been published, with five patients with stable disease (DCR 37.5%) [161].

Studies are investigating the therapeutic effect on enhancing immunity within the TME of Pmmr CRC that may be attained by acting on epigenetic factors such as the bromo- and extraterminal domain (BET). JQ1 is a BET inhibitor that regulates PD-L1 expression, boosts MHC-I mediated cytotoxicity, and downregulates T-reg cell infiltration in TME. Hence, its combination with antiPD-1 exerts a synergic action on CRC cell lines and animal models [162]. The use of other novel agents with atezolizumab is being tested, such as the T-cell bispecific antibody cibisatamab, which acts against carcinoembryonic antigen (CEA) and T-cell CD3. The goal of these types of therapies is to target both a tumor-enriched antigen (such as CEA) and immune cells, and the combination with ICIs may enhance their activity. Cibisatamab has shown promising antitumor activity in preclinical studies by increasing intratumoral T-cell infiltration and upregulating PD-1/PD-L1. In the phase I trial, a partial response was observed in one patient with a Pmmr tumor (10%) [163].

The Inhibition of C-C motif chemokine receptor 5 (CCR5) with maraviroc can polarize macrophages towards the M1 phenotype, which boosts the immune response. Its combination with pembrolizumab in heavily pretreated PmmR tumors was assessed in a phase I trial, with prolonged disease stabilization although a modest 5.3% ORR was observed [164,165].

The combination of ibrutinib and pembrolizumab was tested in a phase I/II trial; ibrutinib is a Bruton’s tyrosine kinase (BTK) inhibitor, which is essential in B-lymphocyte development. This kinase is expressed in up to 90% of CRCs and using ibrutinib in cell lines proved its cytotoxic effect, with a presumed synergistic effect in combination with ICIs. Nonetheless, an ORR of 0% was observed [166].

**Table 11 cancers-15-00863-t011:** Clinical trials involving novel blocking therapies plus ICIs in MSS mCRC.

Study	Treatment	Phase	Endpoint 1	Setting	ORR (%)	MpfS (Months)	MoS (Months)	Status
NCT03821935 [167]	ABBV-151 +/- budigalimab (ABBV-181)	I	Dose escalation and dose expansion	Progression on two prior chemotherapy regimens.	NA	NA	NA	Recruiting
NCT02260440 [158]	Azacitidine + Pembrolizumab	II	ORR	Refractory to standard treatment	3%	1.9	6.3	Completed
CAROSELL (NCT03993626) [159]	Zabadinostat + Nivolumab	I/II	IdcR	Refractory to standard treatments (≥3rd line)	NA	NA	7	Unknown
ENCORE 601 (NCT02437136) [161]	Entinostat + Pembrolizumab MSS McrC cohort	II	MTD and ORR	Refractory to standard treatments	6%	NA	NA	Active, not recruiting
PICCASSO (NCT03274804) [165]	Pembrolizumab + maraviroc	I	Feasibility Rate of a Combined Therapy	Refractory to standard treatments	5.3%	2.1	9.83	Completed
NCT02650713 [163]	Cibisatamab (RO6958688) + atezolizumab	I	DLT, MTD Safety and tolerability	Refractory to standard treatments	10%	NA	NA	Completed
NCT03332498 [166]	Ibrutinib + Pembrolizumab	I/II	DCR 4 months	Refractory to standard treatments	0%	1.4	6.6	Completed

Abbreviations: DLT: dose-limiting toxicity; MTD: maximum tolerable dose; DCR: disease control rate; IdcR: immune disease control rate; RD: recommended dose; ORR: objective response rate; OS: overall survival; PFS: progression-free survival; NR: not reached; NA: not available.

### 3.3. Biomarkers in MSS CRC

There is a strong need to identify new mechanisms that will make MSS tumors sensitive to immunotherapy, in order to achieve the benefits associated with these therapies. Therefore, apart from the combination strategies mentioned, the detection of new biomarkers beyond the MMR status is essential to differentiate the MSS populations for whom development of new strategies using ICIs and other types of immunotherapies may be helpful.

#### 3.3.1. PD-L1

PD-1 is expressed on the surface of T cells, B cells, and natural killer cells and binds to its ligand, PD-L1, which is expressed on the surface of tumor cells to a higher or lower degree. PD-L1 expression or combined positive score (CPS) are used in other malignancies to establish a cutoff from which ICIs may be useful in clinical practice.

In mCRC, regardless of the MMR status, PD-L1 expression has not proved to be predictive of response to ICIs; in the CheckMate 142 clinical trial, a subgroup analysis was performed in the nivolumab group, with no significant differences observed in ORR between PD-L1 <1 and ≥1% (28% vs. 29%, respectively) [168]. In CRC cell lines, observed PD-L1 expression is scarce and below the detection level of IHC techniques, whereas it is predominantly seen in the surround ding myeloid cells; this presentation is unusual in comparison with other immunogenic tumors such as renal or lung cancer, which may explain why PD-L1 is not a trustworthy biomarker for CRC [17].

#### 3.3.2. POLE and POLD1 Mutations

Somatic or germline mutations in the POLE and POLD1 genes are found in 1–2% of all CRC tumors. They lead to alterations in encoding the DNA polymerase epsilon (POLε) and delta (POLδ1), respectively, which play a key role in DNA proofreading and replication. POLε synthesizes the leading strand in the replication process, while POLδ1 oversees the lagging strand. They both enhance the accuracy of replication by recognizing and deleting mismatched base pairs, due in the case of POLε to its exonuclease domain. Therefore, somatic or germline mutations especially in that domain lead to a hypermutated phenotype without dMMR expression [169,170]. Thus, an increase of TILs is found in these tumors, with an average TMB of 158 Mut/Mb, and acquired MMR mutations may develop. In an analysis of 499 CRC tumors, POLE and POLD1 mutated tumors had higher rates of TILs (82%) in comparison with MSI-H (68%) and non-MSI-H CRC (4.5%) [171]

A recent research study reported a greater DCR associated with pathogenic mutations in comparison with the DCR linked to non-actionable variants (82.4% vs. 30.0%; *p* = 0.013). Although most pathogenic mutations are found in the exonuclease domain, others outside this domain have shown similar results in terms of response to ICIs. Increases in mPFS, mOS, and duration of therapy were observed in patients with pathogenic mutations in comparison with those with benign variants, which remained significant when MSI status was considered. Therefore, POLE and POLD1 pathogenic mutations represent a biomarker of response to ICIs regardless of MMR status [172].

In a recent phase II single-arm study, an ORR of 50% was observed in pretreated POLE mutated tumors treated with nivolumab, including MSS CRC [173]. Other studies are currently ongoing to assess the efficacy of nivolumab vs. nivolumab plus ipilimumab in the same setting [174]. In terms of response rate, comparison with MSI status is under evaluation in a phase II study for MSI or POLE mutated mCRC treated with avelumab, with ORR as the primary endpoint [175].

#### 3.3.3. Tumor Mutational Burden

The tumor mutational burden (TMB) refers to the quantity of somatic mutations in a tumor specimen. It is correlated to an elevated neoantigen expression with an associated higher response to ICIs. Its determination is controversial, since it can vary depending on the method used to calculate it. Currently, next-generation sequencing (NGS) techniques are implemented, which differ in their region of study. The process may range from a genome-wide analysis (whole genome sequencing [WGS]) to whole exome sequencing (WES) that includes the entire coding regions and has been considered the reference standard. However, the appearance of large, targeted gene panels such as Foundation One and MSK-IMPACT represents an improvement on prior techniques, and both have been approved by the FDA for profiling solid tumors. Non-MSI tumors with high TMB are estimated to be present in ~3% of cases [176,177,178].

There is no consensus about the TMB cutoff [179] and currently, based on the results from the phase II basket study KEYNOTE-158, pembrolizumab can be used in pre-treated metastatic tumors with TMB ≥ 10 mut/Mb according to the FoundationOne CDx assay, since an ORR of 29.4% was observed with this cutoff vs. 6.3% in TMB < 10 mut/Mb. This correlated with retrospective WES analyses, corresponding to a WES TMB ≥175 mut/exome, in which an ORR of 31.4% was reported (vs. 9.5% when <175 mut/exome) [24,176]. Although mCRC was not included within the spectrum of tumors of this clinical trial, this was not specifically noted in the approval from the FDA [180].

In MyPathway, a phase IIa multibasket study, atezolizumab was tested in advanced solid tumors with TMB ≥10 mut/Mb. The primary endpoint was ORR in patients with TMB ≥16 mut/Mb. A total of 19 tumor types including mCRC were evaluated, with an ORR of 38.1% and DCR of 61.9% vs. 2.1% and 22.9%in TMB <16 mut/Mb, respectively. Focusing on mCRC, 21 patients were assessed, including MSI and MSS tumors, with a better response observed in the former when TMB ≥16 mut/Mb (*n* = 10). An objective response was achieved in three out of five patients with MSS tumors and TMB ≥ 16 mut/Mb. Therefore, TMB may play a role in defining the MSS patients that might respond to ICIs and can also predict patients with MSI mCRC who may not benefit from ICIs [181].

#### 3.3.4. Immunoscore

A subgroup of MSS patients presents a prominent immune gene expression similar to that presented by MSI, leading to higher quantities of CD4+ and CD8+ T cells. These immune infiltrates are associated with better outcomes and a decreased probability of developing metastases. The quantities of cytotoxic (CD8+) and memory (CD3+) T cells are measured in the core and margins of the tumor and can be evaluated using the immunoscore system. Its utility has been proved as a prognostic and response biomarker to ICI; higher immunoscores (I3 or I4) are associated with better disease-specific survival (DSS), which was observed mainly in MSI but also in MSS tumors. Among the MSS tumors analyzed, the risk of relapse in tumors with high Th1 CD4+ and cytotoxic CD8+ T-cell gene expression presented no difference in comparison with MSI but was associated with a significantly lower risk of relapse in comparison with MSS with low gene expression (HR = 2.1; *p* = 0.03) [182,183].

Thus, immunoscore seems a reliable method to predict which patients, including MSS patients, may benefit from immunotherapy, including ICIs and adoptive T-cell therapy, and has already been validated in early-stage (I-III) CRC as a prognostic scale [184].

#### 3.3.5. Microbiome

The human gut microbiome is one of the factors contributing to CRC tumorigenesis, and is formed by the symbiotic microbial cells that form the microbiota. It can be altered by multiple factors that range from genetics to diet or the use of antibiotics. The interaction of the microbiome with the host immunity that determines the homeostasis within the gastrointestinal environment is regulated by microRNAs, which are small non-coding RNAs that participate in post-transcriptional gene regulation. They are secreted by intestinal epithelial cells and released into the lumen, where they modulate the microbiome and establish crosstalk between the microbiome and the host immunity. The interaction is bidirectional, so the gut microbiome may also regulate the expression of microRNAs, as do exogenic dietary microRNAs [185]. Bacterial small RNAs (bsRNAs) also seem to play a role in this crosstalk, although they have been less studied. In a study performed by Tarallo et al., 80 stool specimens from healthy people and patients with CRC or adenomas were collected and analyzed, with higher levers of *Escherichia coli* bsRNAs and DNA in CRC patients in comparison with the other two groups, which may help establish diagnostic models [186].

Interventions to target the gut microbiome can prevent the development of inflammatory or malignant diseases in preclinical models: administration of *Lactobacillus acidophilus* avoided the development of CRC in mice carrying a germline APC mutation [187]. Use of the microbiota to boost the efficacy of treatments is also under investigation: administrating *L. acidophilus* has been demonstrated to modulate tumor growth in preclinical models, reducing oncogenic microRNAs and increasing tumor-suppressor microRNAs in rectal cancer patients [188,189]. The administration of IL-10 and CpG oligodeoxynucleotide proved successful in treating early CRC in mice, due to the release of tumor necrosis factor alpha (TNFα) by *Alistipes shahii* bacterial species, which are over-represented in the colon of mice with CRC. This effect was lost with the use of broad-spectrum antibiotics or germ-free mice.

The effect of local flora on responses to ICIs has been studied in mice, where the use of ipilimumab reduced the growth of tumoral cells in animals with normal microflora. This is believed to depend partly on non-enterotoxin-producing strains of *Bacteroides fragilis*. Curiously, transferring *B. fragilis*-specific CD4+ T cells or transplanting faecal microbiome including Bacteroides species have proved to be effective in reinstating sensitivity to ipilimumab. In cases of anti-PD1 or anti-PDL1, the antitumoral effectiveness is believed to be helped by *Bifidobacterium longum*, *B. adolescentis*, and *B. breve* species. Therefore, the use of a faecal transplant containing the species mentioned may have a role in overcoming the intrinsic resistance in MSS patients [8,190,191,192]. Other species that seem to be favorable for an immune response include *Akkermansia muciniphila*, *Ruminococcaceae,* and *Faecalibacterium*, which were represented in basal samples of patients with superior responses and improved outcomes. The addition of these species by faecal transplant into mice colonized by non-responding patients’ feces showed amelioration of responses to anti-PD(L)1, along with a higher infiltration of immune cells in TME. All the species involved in favorable responses are implicated in decreasing peripheric regulatory T cells and increasing the numbers of dendritic cells and cytotoxic T cells [8,193,194]. Furthermore, the presence of *Lachnospiraceae* and the already mentioned *Ruminococcaceae* have been tested in clinical CRC samples. They were correlated with a higher expression of CCR5 and CXCR3 chemokines, which are involved in the recruitment of T cells and hence may boost the response to ICIs [195].

Currently, clinical trials are testing these hypotheses (NCT04264975, NCT04130763, and NCT05279677), with promising results in refractory to ICIs melanoma patients, since three out of ten refractory patients achieved either PR or CR after faecal transplant [194,196,197,198,199].

In conclusion, further studies are required for to understand how the composition of the microbiota may guide us to predict responses to ICIs and how its modulation may have a role in clinical practice. Faecal transplant might change the immune cell infiltrates and gene expression profiles, which could help establish new treatment strategies for MSS CRC.

## 4. Discussion

CRC represents the third most frequent tumor worldwide and 95% of mCRC cases are MSS [3,6,11]. The use of ICIs has been demonstrated useful in MSI mCRC, with a current indication of pembrolizumab as the standard of care by EMA and FDA, and nivolumab and its combination with ipilimumab by FDA in second or further lines [20,22,29]. However, the use of monotherapy or a combination of ICIs may be less effective in MSS mCRC [134,200].

The immune landscape of mCRC is heterogenous and complex [33]. The differences in TME composition and immune cell infiltrate that exist between MSS and MSI CRC may explain the diverse responses to ICIs observed between these subgroups [201]. The inherent resistance to ICIs in MSS mCRC can be justified by the absence of an effective immune infiltrate for them to work, along with a considerably lower mutation load. Due to these reasons, MSS mCRC tumors require the use of combination strategies and novel agents to increase the antigen load and force the immune system to activate [30,39,202].

The main strategy lies in using combination treatments in order to make these tumors more sensitive to ICIs, either by modulating the immunosuppressant TME and/or by increasing the mutational load. Different strategies are currently under investigation: the combination with CT or RT tries to modulate the surrounding immune infiltrate and obtain from tumoral cell death neoantigens to be presented to the surrounding APC and T cells, which can be enhanced by adding anti-EGFR or anti-VEGF when using CT [203,204]. However, the efficacy of these treatments has not yet been proved, although scant improvements were observed in the combination groups. Furthermore, TMZ can induce a transient MSI status in silenced MGMT MSS mCRC that may bring about sensitization to ICIs [205].

PARP1 is believed to be part of the carcinogenesis of CRC and its role in repairing single-strand DNA damage may be involved in increasing the neoantigen load. Apart from PARP1, other molecules involved in DDR are also being investigated as possible targets to enhance ICIs activity [74,75,76].

Combination strategies with MKI have proved their synergistic effect on other malignancies including renal carcinoma, due to the immune modulator role that these agents present. Although combinations with ICIs have shown better responses than in monotherapy, demonstrating synergistic effects, further studies are required to establish their position in the therapeutic landscape of MSS mCRC [97,98,100,103].

Due to the role of the MAPK pathway in T-cell expansion and immune evasion, MEK inhibitors were first tested in combination with ICIs in these tumors, with unimpressive results. However, promising preliminary results have been observed by targeting BRAF or KRAS when there is a BRAF mutation associated with an MSI phenotype or a KRAS G12C mutation is present, respectively [120,125]. Novel strategies are being assessed including inhibiting DNA methyltransferase or histone deacetylase, which are involved in silencing genes related to the activation of the immune system, with modest responses but considerable DCR achieved [158,159,161].

The combination with other ICIs has not to date provided a clear amelioration of the outcomes presented in monotherapy, although these seem to be improved in cases where no liver metastases are presented or CPS ≥1 [134,135,138,139].

Other signaling pathways involved in the tumorigenesis of MSS CRC include TGF-β and Wnt, due to their roles in promoting angiogenesis and immune evasion in the former and decreasing the recruitment of T cells in the latter. The combination of ICIs with TGF-β inhibitors or monoclonal antibodies has not shown remarkable results so far, but pending clinical trials will clarify their potential role in these tumors [147,148,153].

As previously mentioned, there is a clear necessity of finding robust biomarkers of response beyond the MMR status [206]. PD-L1 expression has not proved its efficacy as a trusty biomarker in CRC, unlike in other malignancies [168]. The analysis of POLE/POLD1 mutations and high TMB might be relevant for finding MSS patients who could benefit from ICIs [173,181]. At the moment, immunoscore seems another promising biomarker, but its utility has only been proved in early-stage CRC [184]. The role that the composition of the microbiome may have in predicting a patient’s responder status, including the possible benefit of using faecal transplant in boosting the response to ICIs in MSS CRC tumors, requires further research [196].

## 5. Conclusions

In conclusion, there is a clear need to improve therapeutic strategies in MSS mCRC to benefit these patients in terms of long-lasting response and the reduced toxicity of ICIs in comparison with standard chemotherapy. ICIs have not yet been approved in this subgroup due to the scarce results presented so far. Hence, combination treatments and the use of novel therapeutic agents are being considered, although modest outcomes have been observed to date. Thus, translational research focused on novel actionable targets and biomarker selection is constantly needed.

## Figures and Tables

**Figure 1 cancers-15-00863-f001:**
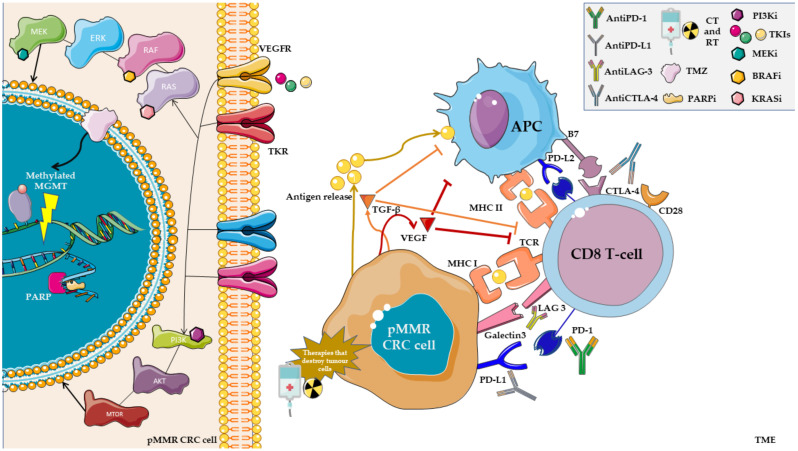
Among the current challenges described in combination with ICIs, we can distinguish the use of therapies that aim to release neoantigens to activate the immune system, such as chemotherapy (CT), targeted therapy, or radiotherapy (RT). Other methods to cope with the inherent resistance to ICIs, such as temozolomide (TMZ) or PARP inhibitors (PARPi), operate by creating a temporary MSI status. The use of tyrosine-kinase inhibitors (TKIs) that target VEGFR in combination with ICIs has had favorable results in other malignancies, with modulation over the TME in addition to the known antitumoral effect. Additionally, blocking TGF- β and VEGF may have a role in promoting ICI activity in these tumors. Meanwhile, there are rationales for combination with inhibitors of the main steps of the MAPK or PIC3KA pathways.

**Table 1 cancers-15-00863-t001:** Clinical trials that evaluated ICIs in MSI mCRC and got the approval from FDA/EMA.

Study	Treatment	Line	Phase	Endpoint 1	ORR (%)	mPFS (Months)	OS
KEYNOTE-016 [20]	Pembrolizumab	After standard treatment based on CT	II	ORR	50%	61% at 24 m	24 m: 66%
KEYNOTE-177 [21]	Pembrolizumab vs. standard treatment	1st line	III	mPFS, mOS	45.1% vs. 33.1%	16.5 vs. 8.2	NR vs. 36.7 m
CheckMate 142 [22]	Nivolumab Nivolumab + ipilimumab	After standard treatment based on CT	II	ORR	31.1% vs. 65%	14.3 vs. NR	NR in both arms

Abbreviations: ORR: Objective response rate, OS: Overall survival, PFS: Progression-free survival, NR: not reached, CT: chemotherapy.

**Table 2 cancers-15-00863-t002:** Currently ongoing and completed clinical trials of anti-VEGF, CT, and ICIs in MSS mCRC.

Study	Treatment	Phase	Endpoint 1	Setting	ORR (%)	mPFS (Months)	mOS (Months)	Status
MODUL (NCT02291289) [45]	Atezolizumab + bevacizumab + fluropyrimidine vs. bevacizumab + fluopyrimidine	II	PFS, OS	BRAF wt pMMR mCRC	NA	7.39 vs. 7.2	21.91 vs. 22.05	Active, not recruiting
BACCI (NCT02873195) [46]	Atezolizumab + capecitabine + bevacizumab vs. capecitabine+bevacizumab	II	PFS	Refractory mCRC	NA	4.4 vs. 3.3	52% vs. 43% (12 m OS)	Active, not recruiting
NCT03396926 [47]	Pembrolizumab + capecitabine + bevacizumab	II	ORR	MSS mCRC	5%	4.3	9.6	Active, not recruiting
CheckMate 9 × 8 (NCT03414983) [48]	Nivolumab + FOLFOX + bevacizumab vs. FOLFOX + bevacizumab	II	PFS	1st line mCRC	60% vs. 46%	11.9 vs. 11.9	29.2 vs. NR	Active, not recruiting
AtezoTRIBE (NCT03721653) [49]	Atezolizumab + FOLFOXIRI + bevacizumab vs. FOLFOXIRI + bevacizumab	II	PFS	1st line mCRC	NA	12.9 vs. 11.4	NA	Active, not recruiting

Abbreviations: DLT: dose-limiting toxicity; MTD: maximum tolerable dose; DCR: disease control rate; iDCR: immune disease control rate; RD: recommended dose; ORR: objective response rate; OS: overall survival; PFS: progression-free survival; NR: not reached; NA: not available.

**Table 4 cancers-15-00863-t004:** Currently ongoing and completed clinical trials of TMZ and ICIs in MSS mCRC.

Scheme 1.	Treatment	Phase	Endpoint 1	Setting	ORR (%)	mPFS (Months)	mOS (Months)	Status
MAYA (NCT03832621) [71]	Nivolumab+ipilimumab+TMZ	II	PFS	MSS MGMT-silenced mCRC	45%	7	18.4	Active, not recruiting
ARETHUSA (NCT03519412) [72]	Pembrolizumab +/- TMZ	II	ORR	Refractory MSI mCRC	NA	NA	NA	Recruiting
NCT04457284 [73]	TMZ + cisplatin + nivolumab	II	ORR	MSS mCRC	NA	NA	NA	Recruiting

Abbreviations: ORR: objective response rate; OS: overall survival; PFS: progression-free survival; NA: not available.

**Table 8 cancers-15-00863-t008:** Currently ongoing and completed clinical trials of combinations with other ICIs in MSS mCRC.

Study	Treatment	Phase	Endpoint 1	Setting	ORR (%)	mPFS (Months)	mOS (Months)	Status
NCT02870920 [134]	Durvalumab + Tremelimumab	II	OS	Refractory to standard treatments	0.8%	1.8 vs. 1.9	6.6 vs. 4.1 MSS: significantly improved	Completed
TAPUR [135] CRC cohort	Nivolumab + ipilimumab High TMB (≥9 Mut/Mb)	II	ORR	Refractory to standard treatments	10%	3.4	10.7	Recruiting (CRC cohort closed due to futility)
NCT03860272 [137]	Botensilimab + Balstilimab	I	DLT Safety and tolerability	Refractory to standard treatments	24%	NA	NA	Recruiting
NCT02720068 [139]	Pembrolizumab + Favezelimab MSS mCRC cohort	I	DLT Safety and tolerability	Refractory to standard treatments (≥3rd line)	6.3%	2.1	8.3	Active, not recruiting

Abbreviations: DLT: dose-limiting toxicity; MTD: maximum tolerable dose; DCR: disease control rate; iDCR: immune disease control rate; RD: recommended dose; ORR: objective response rate; OS: overall survival; PFS: progression-free survival; NR: not reached; NA: not available.

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
