# Peer review of "Current Landscape and Potential Challenges of Immune Checkpoint Inhibitors in Microsatellite Stable Metastatic Colorectal Carcinoma"

_cancers, 2023, doi:10.3390/cancers15030863_

Round 1

Reviewer 1 Report

This review describes mechanisms of resistance to immunotherapy in patients with MSS mCRC and how to overcome them and potential biomarkers. The review is very complete and instructive but need to simplify some parts and the text should be more concise and focused 

Major points:

There is an important therapeutic strategy missing: Inhibition of TGF-beta activity to increase response to immunotherapy. TGF-beta regulates the level and functional activity of numerous immune cell types. Activation of TGF-beta pathway induce an inhibitory effect in immune checkpoints. There are various clinical trials approaching TGF-beta inhibition combined with antiPD1/PDL1 treatments such as M7824 plus 5-FU; Bintrafusp alfa (a bifunctional conjugate that binds TGF-β and PD-L1); or BR102, among others. A chapter dealing with TGF-beta inhibition in order to obtain response to immunological treatments must be included. 

Minor points:

dMMR/MSI-H: simplify to MSI

pMMR/non-MSI-H: simplify to MSS, specifying the first time shown in the text, that in this group MSI-L are included

immunohistochemistry (ICH): IHC instead of ICH 

DNA damage response (DRR): Change to DDR

Relatlimab; add: anti-LAG-3 (ref)

Cobimetinib: add: MEK inhibitor (ref)

Daratumumab: add: anti-CD38 (ref)

Atezolizumab: add: anti-PD-L1 (ref)

Dostarlimab: add: anti-PD1 (ref)

Chapter 2.2 Current treatments approved:

 All the clinical trials information of this chapter can be represented in a table format, facilitating to the readers the comprehension of the clinical trials results.

Chapter 3.2.1. Anti-VEGF and Chemotherapy + Immune Checkpoint Inhibitors:

This chapter described anti-VEGF treatments (bevacizumab) plus chemotherapy and ICI treatments and are represented in table 1 (is not indicated in the text). The text of this chapter needs to be simplified and a final conclusion of all the clinical trials should be added, not a description of each clinical trial, one by one, that is already described in the table. 

The first two paragraphs of this chapter: “CT modulates antitumor immune response………… DCR of 100% at 8 weeks” do not belong to this chapter since they do not approach anti VEGF treatment. 

From Table 1: Anti-VEGF and CT+ICI Trial NCT02375672 (FOLFOX+pembrolizumab) and MEDITREME should not be placed here, since they do not have any anti-VEGF treatment 

Table 1 …..”combinations strategies in pMMR/NON-MSI-H

Just Non-MSI-H? : the first clinical trial NCT02375672 is with dMMR (MSI-H)

I would split table 1 in 7 tables Table 1: Anti_VEGF..; Table 2: Anti-EGFR; Table 3: DDR+ICI;……  

…………Table 7: Novel agents+ICI

3.2.2. Anti-EGFR and Chemotherapy + Immune Checkpoint Inhibitors

Same as in charter 3.2.1, simplified the text and indicate the table. Add a conclusion more than a description of each clinical trial. Table 2 AntiEGFR+CT+ICI, (reference it in the text)

In general reference Table 1; Table 2; Table 3…… in each of the chapters that is not indicated. Table 1 is indicated only once in page 5.

I do not agree with the last paragraph of the discussion, Microbiome modulation is just analyzed mainly in mouse models, is still in basic research, it is not really use in predicting response or in stablishing biomarkers-based strategies.

Author Response

Dear reviewer 1,

Please find attached a point-by-point response your comments.

Kind regards,

María San Román

Reviewer 2 Report

This is a comprehensive review of the use of immune checkpoint inhibitors in metastatic pMMR/MSS colorectal carcinoma.

The sections dealing with the interest and results of clinical trials are detailed and well referenced.

The review is however weaker when dealing with the biology and technology behind the trials. For example:

The paragraph from line 177 to 183 - 

There is, as far as I know, no stage specificity to immune editing, as suggested in the text, it a process that in principal continues throughout tumour development reflecting a ‘Darwinian selection’ process driven by the host immune system. There is no 'specific' selection of mutants with a higher capacity to proliferate and metastasise - only negative selection of mutants recognised by the immune system.

Line 460 - 467 

There are certainly frequent mutations involving the MAPK pathway in tumour cells, but the authors seem to suggest that mutations in this pathway also occurs in reactive T lymphocytes leading to their exhaustion. The reference cited reports work in a mouse model and shows the contradictory effects of MAPK inhibition with loss of activation of naive T cells in lymph nodes but reduced exhaustion of activated T cells. No data is presented as to how this translates in human disease.

Paragraph 710 to 716

There is confusion here between technique NGS (used for exomes, Foundation One and MSK-IMPACT), and panel size, an exome, being simply a big panel. Would need to be re-written.

Line 835 - BRAF mutations do not cause MSI phenotypes, they are associated, but there is no causal relationship

In addition considerable work is required regarding the clarity of the English - for example

Line 62 and 213 ‘o’ for ‘or’

Line 614 unable should be disable

Line 656 Impending need -unclear urgent need???

Line 682 conduit -> lead?

Line711 'correlation to' should be 'surrogate for'

There are many more...

Author Response

Dear reviewer 2,

Please find attached a point-by-point response your comments.

Kind regards,

María San Román

Reviewer 3 Report

This is an interesting review on Current landscape and potential challenges in metastatic colorectal cancer carcinoma

The title should be improved by specifying the meaning of ICI and pMMR (abbreviation not necessary). It is mandatory to be consistent with the use of the abbreviations. 

The introduction should be improved by adding one/two sentences concerning the general molecular biomarkers for CRC

Moreover, please consider the following references (and many others)

Delineating the molecular landscape of different histopathological growth patterns in colorectal cancer liver metastases. Front Immunol. 2022 Dec 16;13:1045329. doi: 10.3389/fimmu.2022.1045329

Mast Cells, microRNAs and Others: The Role of Translational Research on Colorectal Cancer in the Forthcoming Era of Precision Medicine. J Clin Med. 2020 Sep 3;9(9):2852. doi: 10.3390/jcm9092852. PMID: 32899322; PMCID: PMC7564551.

The Evolving Landscape of Immunotherapy in Locally Advanced Rectal Cancer Patients. Cancers (Basel). 2022 Sep 14;14(18):4453. doi: 10.3390/cancers14184453

Why did the authors also consider epigenetic factors? what about radiotherapy?

The concept of microbiota (and epigenetics) is very complex. Authors should improve the dedicated chapter within the review

Novel targets in rectal cancer by considering lncRNA-miRNA-mRNA network in response to Lactobacillus acidophilus consumption: a randomized clinical trial. Sci Rep. 2022 Jun 2;12(1):9168. doi: 10.1038/s41598-022-13297-9

MicroRNAs Regulate Intestinal Immunity and Gut Microbiota for Gastrointestinal Health: A Comprehensive Review. Genes (Basel). 2020 Sep 12;11(9):1075. doi: 10.3390/genes11091075

Altered Fecal Small RNA Profiles in Colorectal Cancer Reflect Gut Microbiome Composition in Stool Samples. mSystems. 2019 Sep 17;4(5):e00289-19. doi: 10.1128/mSystems.00289-19

The authors should emphasize more the clinical implications of the study.

The "biomarkers" chapter is quite confusing and not really helpful

Probably some sections should be merged

Author Response

Dear reviewer 3,

Please find attached the point-by-point response to your comments.

Kind regards,

María San Román

Round 2

Reviewer 1 Report

Authors have improved significantly the original version of the manuscript

Author Response

Dear reviewer 1

Thank you for the revision.

Kind regards.

María San Román and Javier Torres

Reviewer 3 Report

The authors have improved the work.

MSS in the title must be written in full.

Added references should be underlined.

Although the aim of the study is to find "strategy for these patients to benefit from using ICI" the clinical implications need to be highlighted and clarified.

The section on biomarkers remains confusing

Author Response

Dear reviewer 3,

Please find attached a point-by-point response your comments.

Kind regards,

María San Román

Round 3

Reviewer 3 Report

The manuscript can be accepted